# Revealing the Spatial Pattern of Weather-Related Road Traffic Crashes in Slovenia

**Danijel Ivajnšič** [1,2] 🆔, **Nina Horvat** [3], **Igor Žiberna** [2], **Eva Konečnik Kotnik** [2] **and Danijel Davidović** [2,*] 🆔

1 Department for Biology, Faculty of Natural Sciences and Mathematics, University of Maribor, 2000 Maribor, Slovenia; dani.ivajnsic@um.si
2 Department for Geography, Faculty of Arts, University of Maribor, 2000 Maribor, Slovenia; igor.ziberna@um.si (I.Ž.); eva.konecnik@um.si (E.K.K.)
3 Department for Slavic Languages and Literatures, Faculty of Arts, University of Maribor, 2000 Maribor, Slovenia; nina.horvat7@um.si
* Correspondence: danijel.davidovic@um.si

**Abstract:** Despite an improvement in worldwide numbers, road traffic crashes still cause social, psychological, and financial damage and cost most countries 3% of their gross domestic product. However, none of the current commercial or open-source navigation systems contain spatial information about road traffic crash hot spots. By developing an algorithm that can adequately predict such spatial patterns, we can bridge these still existing gaps in road traffic safety. To that end, geographically weighted regression and regression tree models were fitted with five uncorrelated (environmental and socioeconomic) road traffic crash predictor variables. Significant regional differences in adverse weather conditions were identified; Slovenia lies at the conjunction of different climatic zones characterized by differences in weather phenomena, which further modify traffic safety. Thus, more attention to speed limits, safety distance, and other vehicles entering and leaving the system could be expected. In order to further improve road safety and better implement globally sustainable development goals, studies with applicative solutions are urgently needed. Modern vehicle-to-vehicle communication technologies could soon support drivers with real-time traffic data and thus potentially prevent road network crashes.

**Keywords:** GIS; hot spot analysis; traffic safety; spatial modelling; weather patterns

## 1. Introduction

The development of automobilism has made it possible to cross distances more rapidly. Remote, previously inaccessible areas can thus be reached with greater ease. People have better access to jobs, products, and services. Indeed, the improvement of road traffic systems triggers regional, social, and economic development [1]. However, the current fossil fuel-driven road transport has a large carbon footprint [2,3]. It pollutes the environment locally and globally, causes noise and congestion, is a source of additional expense to society, adversely affects people's health, and, in certain circumstances, leads to traffic crashes, some of which cause serious injuries or death [4–7]. These crashes are the eighth leading cause of death for people of all ages and the first leading cause of death for people between the ages of 5 and 29 [8]. Globally, the total number of road deaths in 2016 was 1.35 million (3700 people per day) [8], and in the European Union in 2017, 25,300 people died in road crashes [9].

The incidence of traffic crashes depends mainly on factors such as legislation, road infrastructure, traffic density, roadside surveillance, the condition of motor vehicles, as well as the culture and abilities of drivers and their psychophysical state [10]. In addition to these factors, the weather also has a significant impact. A positive correlation between precipitation and road traffic crash frequency on major roads and highways has been documented in France and on rural roads and motorways in the Netherlands [11]. An

opposite pattern was observed in Athens (Greece), where the number of road traffic crashes was inversely proportional to precipitation [11]. A similar situation was found in Belgium as well, where more traffic crashes occurred in clear and sunny weather, while there were fewer in adverse weather conditions [12]. However, more attention should be paid to road traffic crashes in adverse weather conditions because these usually have a higher death toll [13–15]. At the EU level, in 2016, most crashes occurred in dry weather conditions (70.8%) followed by rain (9.4%), fog (1.4%), snow and hail (0.9%), and strong wind (0.4%) [9]. It has also been found that in rainy, cloudy, and snowy weather, more traffic crashes are caused by women, while in clear weather more crashes are caused by men [16].

Road traffic crashes should receive more attention, especially in Slovenia, since the share of people traveling by car is 86.3% [17], which ranks Slovenia second in the EU. In addition, the number of registered cars in Slovenia is constantly growing (1,165,000 in 2019, which is 2% more than in the previous year) [18]. With more frequent use of cars, the possibility of traffic crashes increases. In fact, the total number of traffic crashes in Slovenia in 2019 increased by 3% compared to 2018, and the number of fatalities increased by 12% [19].

Road safety is one of the fundamental characteristics of transport system quality. The latter is the responsibility of government institutions since they have an overview of transport activities and the means to plan measures. Collection and analysis of data on road traffic crashes in Slovenia is performed within the Sector for the Development and Coordination of Road Safety in the Slovenian Traffic Safety Agency, which operates within the Ministry of Infrastructure. Collecting data is the basis for planning and introducing preventive measures to increase traffic safety. In this context, geographic information system technology enables the implementation of various methods for studying spatial traffic patterns, such as the Bayesian method considering different distribution functions (Poisson, Poisson-gamma, Poisson-lognormal) [20]; hierarchical models [21]; and methods of spatial statistics, such as kernel density estimation [22], local Moran index [23], and the Getis-Ord G* statistics [24]. Moreover, combining machine-learning algorithms and geospatial models can provide applicable solutions to change complex patterns caused by human activity [25,26]. From that perspective, we aimed to figure out the main determinants behind the distinct spatial footprints of road traffic crashes in different weather conditions in Slovenia. In order to do so, we sought answers to the following research questions: (1) Where in Slovenia was the road traffic crash frequency trend (major injury or death) positive or negative in regard to adverse weather conditions? (2) Were there significant hot or cold spots for road traffic crashes on the municipal level? (3) Could the geographically weighted regression approach or machine learning techniques explain the existing spatial pattern of road traffic crashes in different weather conditions?

## 2. Materials and Methods

### 2.1. Databases and Data Preprocessing

National road traffic crash data from 2006 to 2018 were obtained from the Slovenian Traffic Safety Agency web platform [27], which operates under the Ministry of the Interior. Traffic load data for the same time window were downloaded from the Slovenian Open Data Portal (OPSI) owned by the Ministry of Infrastructure [28]. These data are collected at predefined locations but can be interpolated by applying Spatial Analysis along Networks (SANET) tools [29]. Municipal level socioeconomic data (registered personal vehicles, motor vehicles, trailers, trucks, buses, the number of adult persons, regional road length, highway length) were provided by the Statistical Office of the Republic of Slovenia [30] and the Ministry of Finance (municipal development coefficient (MDC)). Climatic and bioclimatic spatial data (resolution = 30 s), version 2.1 (1970–2000), were obtained from the WorldClim database [31]. In order to link all listed data with administrative units (municipalities), vector data were downloaded from the STAGE web application provided by the Statistical Office of the Republic of Slovenia.

The obtained data on road traffic crashes were initially filtered according to their classification (severe injury, death) and/or individual weather situation (rain, snow, wind, fog). In the next step, the standardized road traffic crash rate variable per weather type was calculated [32] since municipalities differ in population density and size:

$$\text{AR\_P} = \text{num. traf. accid.}/\text{tot. pop.} \times 100.000 \tag{1}$$

## 2.2. Trends in Weather-Related Road Traffic Crashes

In order to evaluate regional dynamics in weather-related road traffic crash frequency (severe injury, death), simple municipal-level linear trends were calculated in MS Excel [33]. Here, raw (unstandardized) data from 2006 to 2018 were considered. The total number of road traffic crashes with major/severe injury or death per municipality for each time window and weather condition (rain, snow, wind, fog) was calculated. The linear regression coefficient was then reclassified in three categories in the ArcGIS environment [34]. The first one comprised municipalities with an increasing number of road traffic crashes per weather condition, the second one municipalities with a decreasing number of these events, and the third category with either a constant rate or no road traffic crashes.

## 2.3. Weather-Related Road Traffic Crash Hotspot Analysis

After revealing regional trends in weather-related road traffic crashes, we tried to identify significant spatial clusters of these events. In the first step, we performed a cluster and outlier analysis (Anselin Local Morans I) and continued with the hot spot analysis (Gets-Ord Gi*) by applying the contiguity-edges-corners conceptualization of spatial relationships, since we were operating with polygon features and a standardized dependent variable (AR_P). The Anselin Local Morans I statistics resulted in the following attributes for each municipality: Local Moran's I index, z-score, pseudo *p*-value, and cluster/outlier type. By adding the result of the Gets-Ord Gi* function, statistically significant spatial clusters of high values (AR_P hot spots) and low values (AR_P cold spots) were identified. The tool created an output feature class in the ArcGIS environment with a z-score, *p*-value, and confidence level bin field (Gi_Bin) for each municipality.

## 2.4. Modeling the Weather-Related Traffic Crash Footprint

It is evident that different weather situations create unique spatial patterns of road traffic crashes [12–14]. We sought to find key explanatory variables and develop an algorithm to predict current spatial patterns in Slovenia and potential shifts in the distribution of road traffic crash hot spots. We operated with the dependent variable total frequency of road traffic crashes (severe injury, death) between 2006 and 2018 per municipality and weather situation. Predictor variables (traffic load, registered personal vehicles, motor vehicles, trailers, trucks, buses, the number of adult persons, regional road length, highway length, MDC, climatic and bioclimatic variables) were first tested for multicollinearity (Spearman's correlation coefficient). Redundant variables were removed (VIF > 3). The remaining predictors were then transformed with either factor or principal component analysis (depending on their distributional properties) in the R statistical environment [35] in order to reduce their number and further operate with uncorrelated information.

The following explanatory or predictor variables were transformed with factor analysis: traffic loads, registered personal vehicles, motor vehicles, trailers, trucks, buses, the number of adult persons, regional road length, and highway length. Three factors (explaining 88% of variance) were then used in further modeling procedures. The first factor (F1) loaded information about registered vehicle frequency, population (number of adult persons), and highway length. The second factor (F2) provided information about traffic loads and highway length, and the third factor (F3) loaded mainly information about regional road length.

Principal component analysis was performed on 22 continuous climatic and bioclimatic variables (solar radiation, wind speed, vapor pressure, and 19 bioclimatic variables). All three PCA components (explaining 84% of variability) were then used to calculate

the climate heterogeneity index within the SDM Toolbox for ArcGIS [36]. The climate heterogeneity index (CHI) was considered as the fourth predictor variable in the modelling procedure since we wanted to test the importance and contribution of climatic factors. We completed the list of predictor variables with the MDC variable. This economic development indicator is the ratio between the arithmetic mean of standardized values of indicators in the municipality and the arithmetic mean of standardized values of indicators in the country, where the coefficient of average municipal development in the country equals 1.00.

Finally, two models were calibrated with the dependent variable total frequency of road traffic crashes (severe injury, death) between 2006 and 2018 and the above-mentioned predictors (F1, F2, F3, MDC, and CHI). First, we fitted a geographically weighted regression model (Model Type = Poisson; Spatial Kernel = Adaptive; Bandwidth Searching = Golden Section) with the MGWR 2.2 software (Tempe, AZ, USA) [37] and considered all traffic crashes, regardless of the weather situation. We repeated this procedure for road traffic crashes in rainy, snowy, windy, and foggy conditions. Owing to non-linear relationships between the dependent and independent variables, decision tree models in the R statistical environment were applied in the next step [38,39]. We divided our sample (212 municipalities) for each weather situation into test (25%) and training data (75%). Based on the functional relationships between the dependent and predictor variables in the test data sample, we predicted the number of road traffic crashes in the training data sample for each weather situation. In order to test model quality, standardized residuals were tested for significant spatial autocorrelation with the Moran's I index. However, additional model quality indicators (explained deviance (ED), mean absolute error (MAE), and root mean square error (RMSE)) were also calculated.

## 3. Results

### 3.1. Trends in Road Traffic Crashes in Slovenia

In general, most Slovenian municipalities were characterized by a decreasing trend of road traffic crashes in all considered weather situations except wind. There were 147 municipalities with a negative trend in road traffic crash frequency (70%) and 62 with a positive trend (29.5%) (Figure 1a).

Under rainy weather conditions, the ratio between negative and positive municipal road traffic crash trend coefficients was 135 to 75 (Figure 1b). The positive trend was most prominent in the Ljubljana and Celje Basins, with their surrounding municipalities, and in the most urbanized and the wettest parts of the Dinaric Alps region. The second cluster of municipalities with a positive trend in road traffic crashes in rainy conditions was detected in the eastern part of the Dinaric Alps region. However, during precipitation, the trend was mostly negative in high altitude municipalities with low population density. Similarly, under snowy weather conditions (Figure 1c), the observed trend was negative in 123 municipalities and positive in 77. The spatial footprints of positive or negative trends in road traffic crashes in snowy and rainy conditions were similar. However, more municipalities in the Alps region, which is the coldest and wettest region in Slovenia, had a positive road traffic crash trend in snowy weather conditions. Under strong air advection (wind), positive trends in road traffic crashes with serious injury or death were detected mainly in municipalities in the Mediterranean region, particularly in the Vipava valley (Figure 1d). Here, other plains in the Alps and Pannonian Basin regions were also exposed to turbulent wind conditions, which indirectly caused road traffic crashes between 2006 and 2018. Under dense fog, 115 Slovenian municipalities had a negative and 77 a positive trend in road traffic crash frequency. Basins in the Alps region as well as valleys and karst plateaus in the Dinaric Alps region showed increasing numbers of road traffic crashes in this kind of weather (Figure 1e).

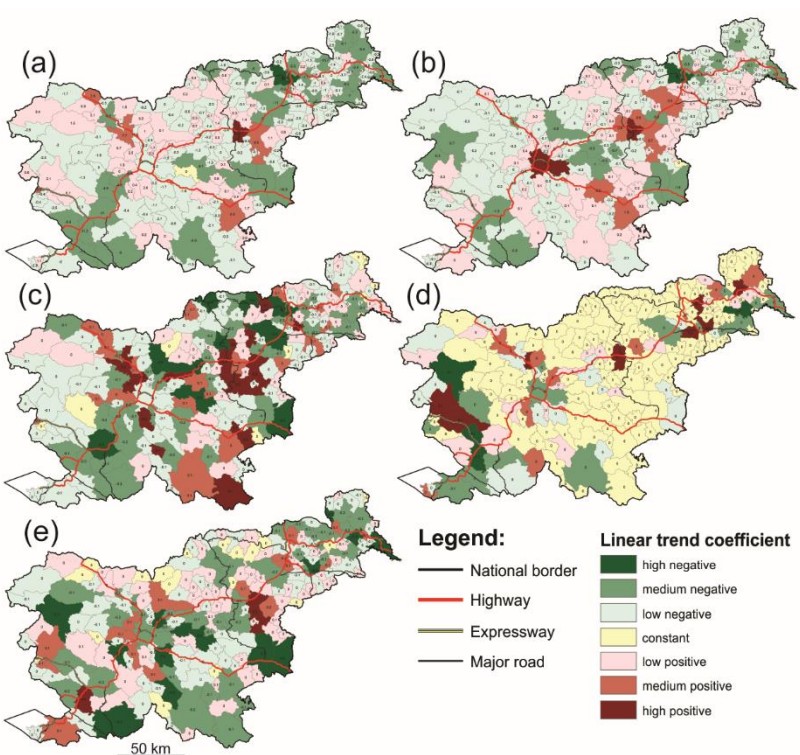

**Figure 1.** Spatial trends in road traffic crashes: (**a**) all crashes (**b**) rain, (**c**) snow, (**d**) wind, (**e**) fog.

### 3.2. Road Traffic Crash Hot Spots in Slovenia

By considering the whole sample (2006–2018) of road traffic crashes in Slovenia (major injury, death), regardless of weather conditions, four significant hot spots emerged (Figure 2a). The largest consisted of municipalities belonging to the Mediterranean region, along the A1 and A3 highways and regional roads connecting the town of Kozina and the Starod border crossing (via Croatia) and the towns of Postojna and Ilirska Bistrica. Two significant road traffic crash hot spots were detected in the hilly parts of the Pannonian Basin region. One comprised three municipalities stretching across the Haloze Hills, and the other one, six municipalities in the Slovenian Hills. In rainy weather conditions, three spatial road traffic crash clusters were detected (Figure 2b). The largest one covered seven municipalities that extend across parts of the Mediterranean and the Dinaric Alps regions. This road crash hot spot was evident in three out of four weather conditions being considered (rain, snow, and wind) (Figure 2b–d). Under snowfall, significant road traffic crash hot spots were identified in municipalities with high relief energy (Phorje (Drava valley), Haloze, Slovenian Hills, Sava Hills, Javorniki Hills and Snežnik Plateau) in different parts of Slovenia. The wind road traffic crash hotspot footprint resulted in only one significant high value cluster. Here, the analysis identified nine municipalities exposed to the turbulent, katabatic, north-eastern Bora wind, which is present in all seasons and reaches the highest speed beneath the high karst plateau. In fog, a large part of the Pannonia Basin region was/is a road traffic crash hot spot (Figure 2e).

### 3.3. The GWR and the Regression Tree Models

In order to properly fit a Poisson GWR model, all predictor variables were tested for possible multicollinearity. Table 1 indicates that all predictors met the basic criteria since correlation coefficients were within the −0.5 and +0.5 margin [40]. However, low variance inflation factors (VIF < 3) additionally excused further use of these predictors in the modelling procedure.

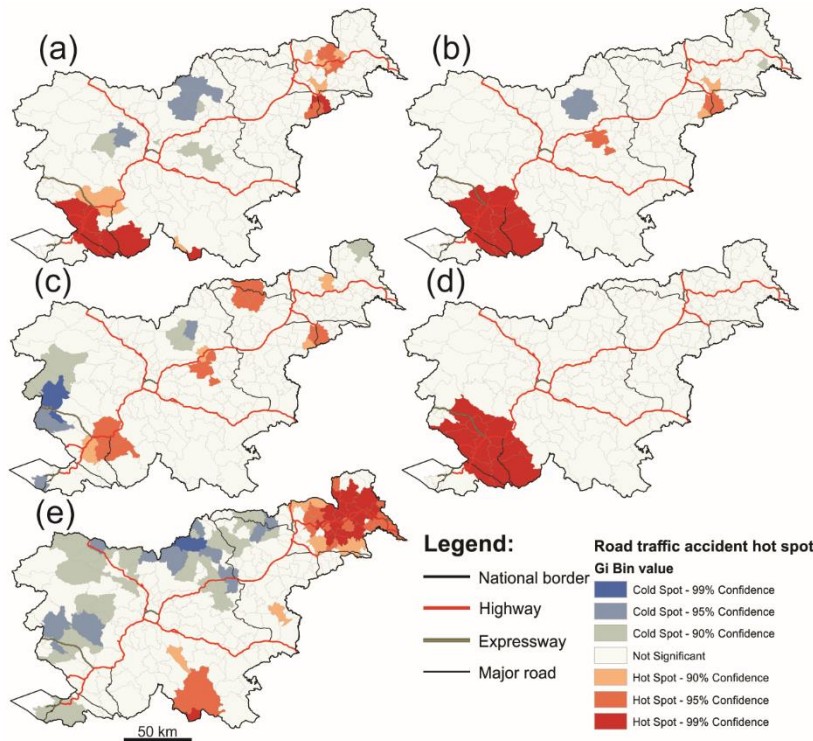

**Figure 2.** Road traffic crash hot and cold spots: (**a**) all crashes, (**b**) rain, (**c**) snow, (**d**) wind, (**e**) fog.

**Table 1.** Predictor variables correlation matrix (Spearman's correlation coefficient).

|  | F1 | F2 | F3 | MDC | CHI |
|---|---|---|---|---|---|
| F1 | 1 | 0.01 | 0.04 | 0.15 | −0.01 |
| F2 | 0.01 | 1 | −0.02 | 0.48 | 0.02 |
| F3 | 0.04 | −0.02 | 1 | 0.15 | 0.18 |
| MDC | 0.15 | 0.48 | 0.15 | 1 | 0.24 |
| CHI | −0.01 | 0.02 | 0.18 | 0.24 | 1 |

F1 = registered vehicle frequency, population (number of adult persons), and highway length; F2 = traffic loads and highway length; F3 = regional road length; MDC = municipality development coefficient; CHI = climate heterogeneity index.

At the initial stage, all road traffic crashes (major injury, death) in the time span 2006–2018, regardless of the weather conditions, were considered as the dependent variable. Global Poisson regression results were produced (Table 2) and then compared against the GWR product (Table 3). All predictor variables had a statistically significant impact on the dependent variable ($p < \alpha$; $\alpha = 0.05$). From the global regression perspective, all predictor estimates (except CHI) had a positive influence on road traffic crash frequency in Slovenia. However, the Monte Carlo spatial variability test indicated that two predictors (F3 and CHI) had significant spatially varying estimates. Moreover, the summary statistics for the GWR parameter estimates (Table 3) showed that predictor F1 (registered vehicle frequency, number of adult persons, and highway length) also belonged to the same category, since its impact on the dependent variable was positive in some municipalities and negative in others.

Despite the relatively high percentage of explained deviance in the global regression (75%), the GWR approach significantly improved model performance. The Ccorrected Akaike information criterion (AICc) was, in this case, 3.7-times lower. However, as soon as we sought to model the weather-related spatial road traffic crash pattern, the GWR approach yielded poor results (maximum explained deviance = 21%). Here, non-linear

relations between the dependent and predictor variables forced us to apply a different methodology to solve this research problem.

**Table 2.** Global regression results, test statistics of predictor variables, and results of the Monte Carlo test for spatial variability.

| | | | | | |
|---|---|---|---|---|---|
| Deviance | 67,674.527 | | | | |
| Log-likelihood | −34,684.57 | | | | |
| AIC | 69,381.143 | | | | |
| AICc | 67,687.532 | | | | |
| Percent deviance explained | 0.755 | | | | |
| Adj. percent deviance explained | 0.749 | | | | |
| Variable | Est. | SE | t (Est/SE) | *p*-value | Spatial Variability *p*-value |
| Intercept | 6.540 | 0.019 | 336.529 | 0.000 | 0.120 |
| F1 | 0.269 | 0.001 | 401.860 | 0.000 | 0.965 |
| F2 | 0.410 | 0.002 | 165.344 | 0.000 | 0.659 |
| F3 | 0.417 | 0.002 | 260.428 | 0.000 | 0.009 |
| MDC | 0.205 | 0.020 | 10.396 | 0.000 | 0.201 |
| CHI | −0.008 | 0.000 | −47.948 | 0.000 | 0.000 |

F1 = registered vehicle frequency, population (number of adult persons), and highway length; F2 = traffic loads and highway length; F3 = regional road length; MDC = municipality development coefficient; CHI = climate heterogeneity index.

**Table 3.** GWR diagnostic information and summary statistics for GWR parameter estimates.

| | | | | | |
|---|---|---|---|---|---|
| Effective number of parameters (trace(S)) | 43.598 | | | | |
| Degree of freedom (n-trace(S)) | 166.402 | | | | |
| Log-likelihood | −10,010.063 | | | | |
| AIC | 18,413.297 | | | | |
| AICc | 18,436.808 | | | | |
| BIC | 18,559.226 | | | | |
| Adj. alfa (95%) | 0.007 | | | | |
| Adj. critical t value (95%) | 2.73 | | | | |
| Variable | Mean | STD | Min | Median | Max |
| Intercept | 5.543 | 1.148 | 3.065 | 5.622 | 8.655 |
| F1 | 0.499 | 0.241 | −0.166 | 0.539 | 1.041 |
| F2 | 0.489 | 0.147 | 0.173 | 0.55 | 0.685 |
| F3 | 0.699 | 0.232 | 0.3 | 0.641 | 1.192 |

**Table 3.** *Cont.*

| | | | | | |
|---|---|---|---|---|---|
| MDC | 0.988 | 1.061 | −1.747 | 1.077 | 3.146 |
| CHI | −0.005 | 0.019 | −0.06 | −0.003 | 0.031 |

F1 = registered vehicle frequency, population (number of adult persons), and highway length; F2 = traffic loads and highway length; F3 = regional road length; MDC = municipality development coefficient; CHI = climate heterogeneity index.

The regression tree models enabled an adequate estimation of road traffic crash frequency in different weather conditions. Functions within the rpart.plot package in the R environment were used to produce Figure 3. The structure of the decision trees indicates the distinct nature of weather-related road traffic crash footprints in Slovenia. In rainy, snowy, and foggy weather conditions, the F3 predictor (length of regional roads) was the main contributor. Climate heterogeneity had a major impact on the spatial pattern of road traffic crashes that appeared under strong air advection (wind). The second branch was more diverse, but CHI played a major role in three out of four weather situations (rain, snow, fog). However, predictors MDC and F2 (traffic load and highway length) were important decision makers as well: the first of these in the spatial road traffic crash footprints under snowy or foggy conditions, and the second one in cyclonic (rain), convective (rain + wind), and advective (wind) synoptic situations.

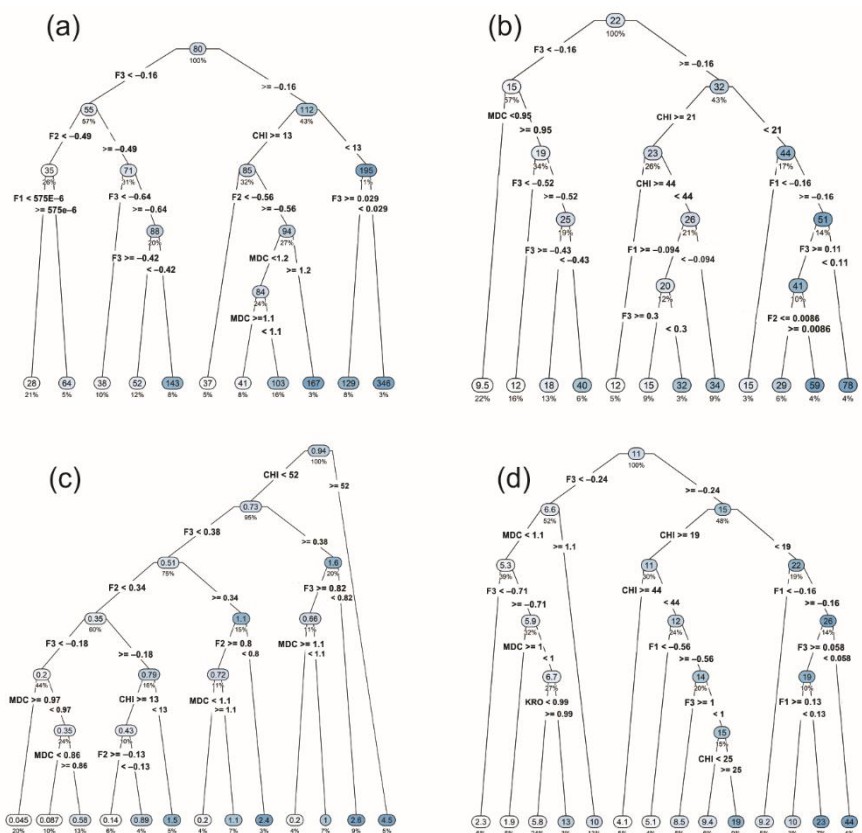

**Figure 3.** Regression trees for road traffic crashes in (**a**) rain (**b**) snow, (**c**) wind, and (**d**) fog.

However, Table 4 summarizes the overall predictor variable importance per weather situation. We reached a much higher explained deviance parameter with the regression tree models compared to that of the GWR. The mean absolute error value estimates how well the models predicted our dependent variables (road traffic crash frequency by weather situation) on a yearly level. The best result was produced for the wind-related road traffic crash pattern. Model over- and under-predictions for all weather situations, represented with standardized residuals, were all normally distributed and free of spatial

autocorrelation (insignificant Moran's I values). Of course, despite the properly specified regression tree models, results could probably be further improved if additional linearly related predictors were considered.

**Table 4.** Summary statistics for the decision tree models.

| Weather Situation | Variable | Variable Importance (%) | Deviance Explained (%) | MAE | Moran's Index | *p*-Value |
|---|---|---|---|---|---|---|
| Rain | F3 | 36 | 43 | 4.33 | −0.047 | 0.816 |
| | F1 | 17 | | | | |
| | CHI | 17 | | | | |
| | F2 | 16 | | | | |
| | MDC | 15 | | | | |
| Snow | F3 | 39 | 41 | 1.23 | −0.142 | 0.472 |
| | F1 | 21 | | | | |
| | CHI | 17 | | | | |
| | F2 | 12 | | | | |
| | MDC | 11 | | | | |
| Wind | CHI | 32 | 48 | 0.08 | 0.157 | 0.376 |
| | F3 | 25 | | | | |
| | MDC | 16 | | | | |
| | F2 | 15 | | | | |
| | F1 | 12 | | | | |
| Fog | F3 | 32 | 43 | 0.58 | −0.091 | 0.635 |
| | CHI | 20 | | | | |
| | MDC | 18 | | | | |
| | F1 | 17 | | | | |
| | F2 | 13 | | | | |

## 4. Discussion

Owing to decreased traveling time and increased mobility, motor vehicles provide many benefits for individuals and society [1]. However, their widespread use is negatively affecting the quality of the living environment and poses a threat in the form of traffic crashes, which are still the leading cause of death for children and young adults [41,42]. The increased use of vehicles and more frequent extreme weather events, because of climate change, can thus result in higher road traffic crash risk, despite an improvement in worldwide numbers [43].

The Slovenian road traffic crash database revealed a similar frequency distribution of road traffic crashes in adverse weather conditions as reported in other European countries. Fortunately, general trends for these unpleasant events are, on both levels (national and European), clearly negative [44]. However, Romano and Jiang [45] concluded that road traffic crashes are spatiotemporal events along road networks, which can be influenced by many varying factors, including weather. Because Slovenia lies at the conjunction of different climatic zones characterized by differences in weather phenomena, which further modify traffic safety, significant regional differences in road traffic crash frequency were expected. Other studies across the Globe [11,12,43,44,46,47] have confirmed the impact of weather (as a direct or indirect cause) on road traffic crashes, especially during heavy precipitation. Slovenia is not an exception; rainy and snowy weather conditions lie behind the highest share of road traffic crashes in adverse weather. The municipal level road traffic crash trend analysis, based on 13 time windows (2006–2018), additionally confirmed this fact. However, the road traffic crash hot spot pattern in Slovenia was the most unusual in windy and foggy weather conditions. Moreover, this unique hot spot pattern was detected in the four Slovenian municipalities with the highest traffic load and crash frequency (Ljubljana, Maribor, Celje, and Koper) on the local road segment level [48].

Weather conditions—despite the great natural and geographical diversity of Slovenia—are not the main cause of traffic crashes, but they are an important modifier, especially in areas with relief diversity. Particularly problematic are areas with narrow valleys, where pools of cold air appear at the bottom, which are associated with more a frequent occurrence of ice or fog. If we add to this the fact that in some areas these roads are the only connections between regional centers so the traffic density is higher (the Drava and Soča valleys), the consequence is a higher probability of traffic crashes.

By developing an algorithm to adequately predict spatial patterns of anthropogenic activity, here manifested in the form of road traffic crashes, we can bridge still existing gaps in road traffic safety. There is no doubt that results from geospatial models are usually highly applicable [49], but none of the current commercial or open-source navigation systems contain spatial information about road traffic crash hot spots. Thus, more attention to speed limits, safety distance, and other vehicles entering and leaving the system could be expected. Similar to the Google Maps navigation system app, which provides real-time information about traffic load on highways, our findings could be integrated into a spatial decision support system that warns drivers who are entering or leaving any municipal road network system with higher road traffic crash risk depending on the given weather conditions. Ivajnšič et al. [48] developed such an app for the android environment, but for only four Slovenian municipalities. They linked the same road traffic crash dataset with road network vector data and thus identified more (death and major injury) and less (minor injury and material damage) dangerous road segments in different weather conditions. These findings were later transferred into the SLOCrashInfo mobile app where dangerous road segments are displayed as visual warnings (death and major injury = red screen alert, minor injury and material damage = blue screen alert, etc.) on the OSM basemap. Moreover, this app also functions as a navigation system with which dangerous road segments can be avoided. However, in order to properly raise drivers' awareness about road traffic crash hot spots, such spatial information should be provided on the European (or even international) level. In that case, data availability, data capacity, and computer processing power are the main limiting factors for now. Nonetheless, some authors [50,51] have emphasized that modern vehicle-to-vehicle communication technologies could support drivers with real-time traffic data and thus potentially prevent road network crashes. The ideal solution would be integrating such informative spatial data with the vehicle information system. The forthcoming internet of things (IOT) platform in the transport sector provides a good opportunity for the development of decision support systems for road traffic safety.

Our findings could also be linked with the signaling systems along highways. The interactive information signs could project this kind of information and thus inform drivers about dangerous highway segments according to the given weather conditions. However, in this case, regional and local roads, which are not supported with such signaling technology, would be left behind.

Because the social, psychological, and financial damage caused by road crashes worldwide is still enormous and road traffic crashes cost most countries 3% of their gross domestic product [44], studies like this one that provide applicative solutions to potentially increase road traffic safety are urgently needed.

**Author Contributions:** D.I.: Supervision, Conceptualization, Methodology, Formal analysis, Writing—review and editing. N.H.: Data curation, Formal analysis. E.K.K.: Data curation, Writing—original draft. I.Ž.: Visualization, Investigation, Formal analysis. D.D.: Formal analysis, Writing—review and editing. All authors have read and agreed to the published version of the manuscript.

**Funding:** This study was supported by the Slovenian Research Agency and the Research Program Slovene identity and cultural awareness in linguistic and ethnic contact areas in past and present (P6-0372), the research project Preventing heat stress in urban systems (J7-1822) and the project »Development Of Research Infrastructure For The International Competitiveness Of The Slovenian RRI Space—RI-SI-LifeWatch« co-financed by the Republic of Slovenia, Ministry of Education, Science and Sport and the European Union from the European Regional Development Fund.

**Institutional Review Board Statement:** Not applicable.

**Informed Consent Statement:** Not applicable.

**Data Availability Statement:** Data available on request at danijel.davidovic@um.si (accessed on 11 March 2021).

**Conflicts of Interest:** The authors declare no conflict of interest.

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
