# Peer review of "Revealing the Spatial Pattern of Weather-Related Road Traffic Crashes in Slovenia"

_applsci, doi:10.3390/app11146506_

Round 1

Reviewer 1 Report

The authors asked themselves the following questions:

  1. Where in Slovenia was the road traffic accident frequency trend positive or negative regarding adverse weather conditions?
  2. Were there significant hot or cold spots for road traffic accidents on the municipal level?
  3. Could the geographically weighted regression approach or machine learning techniques explain the existing spatial pattern of road traffic accidents in different weather conditions?

Using the data available to them (national road traffic accident data from 2006 to 2018; traffic load data; municipal level socio-economic data such as registered personal vehicles, motor vehicles, trailers, trucks, buses, the number of adult persons, regional road length, and highway length; climatic and bioclimatic spatial data), they:

  • linked all listed data;
  • determined trends in weather-related road traffic accidents;
  • conducted weather-related road traffic accident hotspot analysis
  • conducted modeling the weather-related traffic accident footprint.

Finally, which is very important, they emphasized the contribution of their analysis as well as how the results of this analysis could be used to improve traffic safety.

Author Response

The authors asked themselves the following questions:

  1. Where in Slovenia was the road traffic accident frequency trend positive or negative regarding adverse weather conditions?
  2. Were there significant hot or cold spots for road traffic accidents on the municipal level?
  3. Could the geographically weighted regression approach or machine learning techniques explain the existing spatial pattern of road traffic accidents in different weather conditions?

Using the data available to them (national road traffic accident data from 2006 to 2018; traffic load data; municipal level socio-economic data such as registered personal vehicles, motor vehicles, trailers, trucks, buses, the number of adult persons, regional road length, and highway length; climatic and bioclimatic spatial data), they:

  • linked all listed data;
  • determined trends in weather-related road traffic accidents;
  • conducted weather-related road traffic accident hotspot analysis
  • conducted modeling the weather-related traffic accident footprint.

Finally, which is very important, they emphasized the contribution of their analysis as well as how the results of this analysis could be used to improve traffic safety.

RESPONSE: Thank you very much for your positive opinion regarding our paper.

Reviewer 2 Report

I commend the authors for this original manuscript that tackles the spatial patterns of weather-related traffic crashes in Slovenia. The manuscript is well structured and well written, which greatly facilitates following and understanding of the arguments. The research questions are well outlined and help clarify what to expect from the remainder of the manuscript. To improve the manuscript, I have a few suggestions (minor), which I provide in the form of general and specific comments as follows:

General comments:

I think the manuscript needs a firmer conclusion section that explains what the findings mean in terms of policy. A few suggestions have been provided (e.g., using interactive information signs to inform drivers of dangerous road segments). However, it is not clear how the findings (which are based on municipal-level data) can be generalized/applied to other settings such as local/rural roads. Perhaps a few sentences can be added to discuss the transferability/applicability of the findings to other roadway settings (for instance, rural roads) and also, what the policy recommendations that would entail.

Specific comments:

Lines 33 - 36: In the sentences that start with “It pollutes..”, please provide a few references for the arguments being made.

Line 35 (and also throughout the manuscript): Please consider using the word “crashes” instead of the word “accidents” in reference to traffic collisions. There are plenty of research papers (at least over the past two decades) on the topic of safety that recommend using the term “crash” instead of “accident” (when referring to traffic collisions) with explanations of reasons. Please refer to the below example:

Stewart, A.E. and Lord, J.H., 2002. Motor vehicle crash versus accident: a change in terminology is necessary. Journal of traumatic stress, 15(4), pp.333-335.

URL: https://link.springer.com/article/10.1023/A:1016260130224

Line 52: Please insert a comma after the phrase “in 2016” to make it read “At the EU level, in 2016, most accidents…”

Lines 70 and 77: Please consider changing the phrase “From that perspective,” in one of these lines because it is repeated too closely to each other within the manuscript.

Line 91: The word “data” is a plural word (as referred to correctly in Lines 90, 95, and 97, and more lines by using the verb “were” in reference to the word “data”). Please modify the sentence starting with “This data is” to “These data are” to reflect correct grammar.

Line 232: Please provide a reference for choosing the -0.5 and +0.5 margins as the selection criteria for the predictors based on their correlation coefficient. I know this is a very common practice in research; however, adding a reference for such arguments would add to the strength of your paper.

Author Response

I commend the authors for this original manuscript that tackles the spatial patterns of weather-related traffic crashes in Slovenia. The manuscript is well structured and well written, which greatly facilitates following and understanding of the arguments. The research questions are well outlined and help clarify what to expect from the remainder of the manuscript. To improve the manuscript, I have a few suggestions (minor), which I provide in the form of general and specific comments as follows:

General comments:

I think the manuscript needs a firmer conclusion section that explains what the findings mean in terms of policy. A few suggestions have been provided (e.g., using interactive information signs to inform drivers of dangerous road segments). However, it is not clear how the findings (which are based on municipal-level data) can be generalized/applied to other settings such as local/rural roads. Perhaps a few sentences can be added to discuss the transferability/applicability of the findings to other roadway settings (for instance, rural roads) and also, what the policy recommendations that would entail.

RESPONSE: thank you for your constructive comments. We agree with the proposed suggestions and have updated the manuscript (please see the attachment). The Discussion section was improved. The applicability of the developed results is now better discussed (now lines 340-363)

Specific comments:

Lines 33 - 36: In the sentences that start with “It pollutes..”, please provide a few references for the arguments being made.

RESPONSE: Done. Now 4 references are provided.

Line 35 (and also throughout the manuscript): Please consider using the word “crashes” instead of the word “accidents” in reference to traffic collisions. There are plenty of research papers (at least over the past two decades) on the topic of safety that recommend using the term “crash” instead of “accident” (when referring to traffic collisions) with explanations of reasons. Please refer to the below example:

Stewart, A.E. and Lord, J.H., 2002. Motor vehicle crash versus accident: a change in terminology is necessary. Journal of traumatic stress, 15(4), pp.333-335.

URL: https://link.springer.com/article/10.1023/A:1016260130224

RESPONSE: Done throughout the manuscript, title including.

Line 52: Please insert a comma after the phrase “in 2016” to make it read “At the EU level, in 2016, most accidents…”

RESPONSE: Done.

Lines 70 and 77: Please consider changing the phrase “From that perspective,” in one of these lines because it is repeated too closely to each other within the manuscript.

RESPONSE: Done. Phrase changed.

Line 91: The word “data” is a plural word (as referred to correctly in Lines 90, 95, and 97, and more lines by using the verb “were” in reference to the word “data”). Please modify the sentence starting with “This data is” to “These data are” to reflect correct grammar.

RESPONSE: Done. Grammar modified.

Line 232: Please provide a reference for choosing the -0.5 and +0.5 margins as the selection criteria for the predictors based on their correlation coefficient. I know this is a very common practice in research; however, adding a reference for such arguments would add to the strength of your paper.

RESPONSE: Done. Reference 40 added.

Sincerely,

Danijel Davidovič, corresponding author
